# A Comparative Study of Swarm Intelligence Metaheuristics in UKF-Based Neural Training Applied to the Identification and Control of Robotic Manipulator

Juan F. Guerra, Ramon Garcia-Hernandez *, Miguel A. Llama and Victor Santibañez

Tecnologico Nacional de Mexico, Instituto Tecnologico de La Laguna, Torreon 27000, Mexico;
m.jfguerrac@correo.itlalaguna.edu.mx (J.F.G.); mllama@lalaguna.tecnm.mx (M.A.L.);
vasantibanezd@lalaguna.tecnm.mx (V.S.)
* Correspondence: rgarciah@lalaguna.tecnm.mx

**Abstract:** This work presents a comprehensive comparative analysis of four prominent swarm intelligence (SI) optimization algorithms: Ant Lion Optimizer (ALO), Bat Algorithm (BA), Grey Wolf Optimizer (GWO), and Moth Flame Optimization (MFO). When compared under the same conditions with other SI algorithms, the Particle Swarm Optimization (PSO) stands out. First, the Unscented Kalman Filter (UKF) parameters to be optimized are selected, and then each SI optimization algorithm is executed within an off-line simulation. Once the UKF initialization parameters $P_0$, $Q_0$, and $R_0$ are obtained, they are applied in real-time in the decentralized neural block control (DNBC) scheme for the trajectory tracking task of a 2-DOF robot manipulator. Finally, the results are compared according to the criteria performance evaluation using each algorithm, along with CPU cost.

**Keywords:** swarm intelligence; neural networks; robot control; unscented Kalman filter





## 1. Introduction

Metaheuristics can be classified into various categories based on their natural inspiration [1]. One prominent category is swarm intelligence-based algorithms, which draw inspiration from the collective behavior of social insect colonies, bird flocks, or animal herds. Swarm intelligence (SI) algorithms simulate the cooperative and self-organizing behavior observed in natural swarms to solve complex optimization problems [2,3].

SI, inspired by the collective behavior of social insect colonies, encompasses a diverse range of algorithms that facilitate efficient problem-solving through cooperation and self-organization. These algorithms simulate the collaboration and information exchange observed in natural swarms, enabling them to achieve global optimization. By harnessing the collective intelligence exhibited by swarm systems, SI metaheuristics offer promising avenues for optimizing neural network training and enhancing the identification and control capabilities of robotic systems.

To conduct a comprehensive analysis, we selected four state-of-the-art SI algorithms known for their unique characteristics and optimization strategies. Ant Lion Optimization (ALO), drawing inspiration from the hunting behavior of ant lions; employs a powerful search mechanism to explore and exploit the solution space efficiently. Bat Algorithm (BA) mimics the echolocation behavior of bats, utilizing frequency tuning and pulse emission concepts to achieve effective optimization. Grey Wolf Optimizer (GWO) emulates the social hierarchy and hunting dynamics of grey wolves, employing three fundamental types of wolf-inspired operators to strike a balance between exploration and exploitation. Moth Flame Optimization (MFO), inspired by the moth's phototaxis behavior toward flames, incorporates attraction and repulsion mechanisms to guide the optimization process effectively. Finally, we compare these algorithms against the well-established Particle Swarm Optimization (PSO), which draws inspiration from the social behavior of bird

flocking, enabling particles to adaptively search the solution space based on individual and swarm experience.

Overall, SI algorithms offer powerful optimization techniques that leverage the collective intelligence and self-organization observed in natural swarms. Their robustness, global exploration capabilities, self-adaptation, parallelism, scalability, and bio-inspired concepts make them well-suited for addressing a wide range of optimization problems in various domains. By mimicking the behavior of swarms, these algorithms provide effective solutions and insights for solving complex optimization challenges [4,5].

Furthermore, the nature-inspired characteristics of SI optimization methods introduce robustness and adaptability to different problem domains. They can be readily applied to robotic systems, including those with complex dynamics and uncertain environments [6]. By employing metaheuristic optimization techniques, the Unscented Kalman Filter (UKF) initialization parameters can be tailored to specific robotic platforms and tasks, leading to an improved estimation and control performance.

By integrating SI algorithms with UKF-based neural training, we aim to improve the accuracy of identification and control in a two-degrees-of-freedom (DOF) robot manipulator.

To elaborate further, let us delve into the distinguishing features and underlying principles of ALO, BA, GWO, and MFO algorithms. ALO utilizes a population of artificial ant lions to mimic hunting behaviors, where each ant lion represents a potential solution in the search space [7]. The algorithm employs pride update and position update mechanisms to perform efficient exploration and exploitation. BA, on the other hand, emulates the echolocation behavior of bats to optimize solutions. Bats navigate through a combination of random flight, frequency tuning, and pulse emission, allowing them to find optimal solutions in dynamic environments [8]. GWO, inspired by the cooperative hunting dynamics of grey wolves, utilizes three types of wolf operators (alpha, beta, and delta) to balance exploration and exploitation. The alpha wolf coordinates exploration, while the beta and delta wolves perform local exploitation and global exploration, respectively [9]. MFO draws inspiration from the attraction of moths to flame, employing attraction and repulsion mechanisms to guide the optimization process effectively. Moths are attracted to the light source but are also repelled by other moths, leading to a balanced exploration–exploitation trade-off [10].

To compare these algorithms against the widely used PSO, we consider PSO ability to adaptively search the solution space based on individual and swarm experience. PSO employs velocity updates and position adjustments to explore and exploit the search space efficiently [1,11,12]. By contrasting the unique characteristics and optimization strategies of ALO, BA, GWO, and MFO with the PSO algorithm, we can gain valuable insights into their relative strengths and weaknesses by applying them to our proposed identification and control scheme. We selected these algorithms mainly because of the following advantages: their small numbers of tuning parameters, low CPU time costs, the ability to maintain joint torque limits, and a better overall performance than other SI algorithms, such as Artificial Bee Colony (ABC), Ant Colony Optimization (ACO), Cuckoo Search (CS), Accelerated Particle Swarm Optimization (APSO), and Whale Optimization Algorithm (WOA).

The structure of this work is as follows. In Section 2, we mentioned some main characteristics of the SI algorithms employed. Section 3 describes the methodology of neural identification and control scheme. The simulation and real-time results for trajectory tracking are presented in Section 4. Discussions of the results are reflected in Section 5. Finally, concluding remarks are given in Section 6.

## 2. Swarm Intelligence Algorithms

Metaheuristics are a family of optimization algorithms designed to find suitable solutions for complicated optimization problems. In contrast to traditional optimization methods, which aim to find the global optimal, metaheuristic algorithms obtain acceptable results quickly, even in the presence of multiple local optima.

In summary, metaheuristics are fantastic tools for finding good solutions to a wide variety of optimization problems. They are especially useful in situations where traditional methods are not effective, such as problems with high dimensionality, non-convex, noise, or incomplete data.

Bio-inspired algorithms are unique metaheuristic methods inspired by natural processes, phenomena, concepts, and systems mechanisms. Each has features and strengths that provide interpretability and inspiration for solving real-world problems in diverse fields, such as engineering, computer science, economics, and biology. These algorithms mimic the behavior of systems in nature, such as evolutionary computation and swarm behavior [1].

SI is a subfield of bio-inspired algorithms that draws inspiration from collective behavior in nature and focuses on the emergent behavior of decentralized populations through local interactions and self-organization. Figure 1 shows the flow chart of the proposed methodology, which is described as follows: firstly, the UKF parameters to be optimized are selected; then, each SI optimization algorithm is executed within an off-line simulation. Once the UKF initialization parameters $P_0$, $Q_0$, and $R_0$ are obtained, they are applied in real-time in the decentralized neural block control (DNBC) scheme for the trajectory tracking task of a 2-DOF robot manipulator. Finally, the results are compared according to the objective function evaluation.

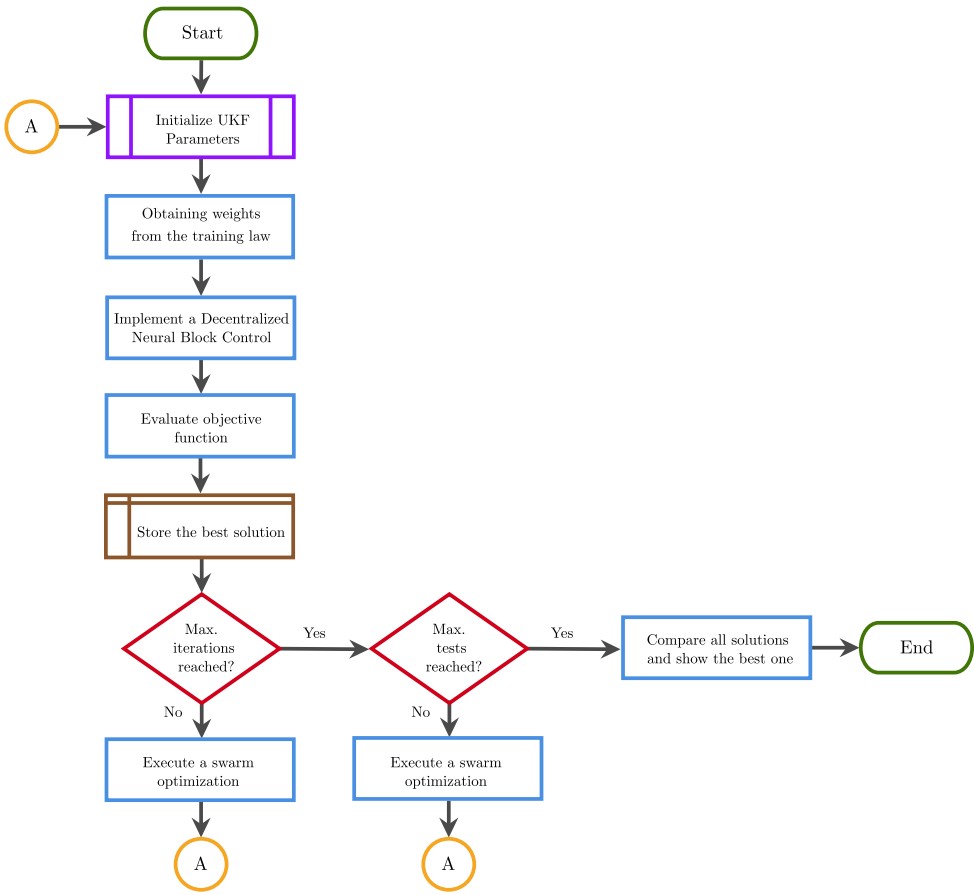

**Figure 1.** Proposed methodology flowchart.

Although there is an endless number of SI algorithms, which have multiple modifications, in the proposed approach, we use algorithms in their original versions, of which a brief description is presented below.

### 2.1. Particle Swarm Optimization (PSO)

Particle Swarm Optimization (PSO) is a widely used metaheuristic algorithm inspired by the collective behavior of bird flocks or fish schools. It has demonstrated remarkable success in solving various optimization problems [11]. PSO operates on the principle of iteratively adjusting the positions and velocities of particles in a multidimensional search space. Algorithm 1 shows the implementation of PSO [12].

---

**Algorithm 1** PSO Pseudocode

---

**Require:**
1: $n$: number of particles
2: $d$: dimension of the search space
3: $t_{max}$: maximum number of iterations
4: $w$: inertia weight
5: $\varphi_1$: cognitive acceleration coefficient
6: $\varphi_2$: social acceleration coefficient
7: $x_i$: position of particle $i$
8: $v_i$: velocity of particle $i$
9: $pbest_i$: best position of particle $i$
10: $gbest$: global best position
**Ensure:**
11: $x^*$: optimal solution
12: $f^*$: optimal fitness
13: **Function** PSO( )
14:     **for** $i = 1$ to $n$ **do**
15:         Initialize $x_i$ and $v_i$ randomly
16:     **end for**
17:     **while** $t < t_{max}$ **do**
18:         **for** $i = 1$ to $n$ **do**
19:             Update velocity: $v_i \leftarrow wv_i + \varphi_1(pbest_i - x_i) + \varphi_2(gbest - x_i)$
20:             Update position: $x_i \leftarrow x_i + v_i$
21:         **end for**
22:         **for** $i = 1$ to $n$ **do**
23:             **if** $f(x_i) < f(pbest_i)$ **then**
24:                 $pbest_i \leftarrow x_i$
25:             **end if**
26:             **if** $f(x_i) < f(gbest)$ **then**
27:                 $gbest \leftarrow x_i$
28:             **end if**
29:         **end for**
30:         $t \leftarrow t + 1$
31:     **end while**
32:     $x^* \leftarrow gbest$
33:     $f^* \leftarrow f(gbest)$
34: **end Function**

---

### 2.2. Ant Lion Optimizer (ALO)

The Ant Lion Optimizer (ALO) is a potent metaheuristic algorithm inspired by the predatory behavior of ant lions. It has gained significant attention in optimization engineering due to its ability to deal with complex problems effectively. ALO displays unique characteristics that distinguish it from other metaheuristics, such as PSO [7].

ALO emulates the hunting strategy employed by ant lions to capture their prey, which consists of building conical pits in sandy areas. For optimization problems, ALO capitalizes on this natural behavior to explore and exploit the solution space efficiently. Algorithm 2 shows the implementation of ALO [13].

---

**Algorithm 2** ALO Pseudocode

---

**Require:**

  1:   $p$: number of antlions

  2:   $u$: upper bounds of variables

  3:   $l$: lower bounds of variables

  4:   $alpha$: evaporation rate

  5:   $beta$: attractiveness rate

  6:   $t_{max}$: maximum number of iterations

  7:   $x_i$: position of antlion $i$

  8:   $f(x_i)$: fitness of antlion $i$

  9:   $x_i^*$: best position of antlion $i$

10:   $f(x_i^*)$: best performance of antlion $i$

**Ensure:**

11:   $x^*$: optimal solution

12:   $f^*$: optimal fitness

13:   **Function** ALO( )

14:     **for** $i = 1$ to $p$ **do**

15:       Initialize $x_i$ randomly

16:     **end for**

17:     **while** $t < t_{max}$ **do**

18:       **for** $i = 1$ to $p$ **do**

19:         Generate a new position $x_i'$

20:         Calculate the fitness of $x_i'$

21:         **if** $f(x_i') < f(x_i)$ **then**

22:           $x_i \leftarrow x_i'$

23:         **end if**

24:       **end for**

25:       **for** $i = 1$ to $p$ **do**

26:         Evaporation: $x_i \leftarrow x_i + \alpha(x_i^* - x_i)$

27:         Attractiveness: $x_i \leftarrow x_i + \beta(x_i - x_{best})$

28:       **end for**

29:       $t \leftarrow t + 1$

30:     **end while**

31:     $x^* \leftarrow x_{best}$

32:     $f^* \leftarrow f_{best}$

33:   **end Function**

---

### 2.3. Bat Algorithm (BA)

The Bat Algorithm (BA) is an SI algorithm inspired by the echolocation behavior of bats. The BA demonstrates unique characteristics that set it apart from other metaheuristic algorithms [8].

The algorithm begins by initializing a population of bats, where each bat represents a potential solution to the optimization problem. Bats fly through the search space, continuously adjusting their positions and velocities based on their knowledge. BA implementation is illustrated in Algorithm 3 [1].

---

**Algorithm 3** BA Pseudocode

---

**Require:**
1: $n$: number of bats
2: $d$: dimension of the search space
3: $t_{max}$: maximum number of iterations
4: $A$: loudness
5: $r$: pulse rate
6: $\alpha$: cooling factor
7: $\gamma$: wavelength
8: $x_i$: position of bat $i$
9: $f(x_i)$: fitness of bat $i$
10: $x_i^*$: best position of bat $i$
11: $f(x_i^*)$: best fitness of bat $i$
**Ensure:**
12: $x^*$: optimal solution
13: **Function** BA()
14:    **for** $i = 1$ to $n$ **do**
15:       Initialize $x_i$ randomly
16:    **end for**
17:    **while** $t < t_{max}$ **do**
18:       **for** $i = 1$ to $n$ **do**
19:          Generate a new position $x_i'$
20:          Calculate the fitness of $x_i'$
21:          **if** $f(x_i') < f(x_i)$ **then**
22:             $x_i \leftarrow x_i'$
23:          **end if**
24:          Update loudness: $\alpha_i \leftarrow \alpha_i - 1$
25:          Update pulse rate: $\beta_i \leftarrow \beta_i + 1$
26:       **end for**
27:       **for** $i = 1$ to $n$ **do**
28:          Probability of loudness: $p_i = \frac{1}{\alpha_i}$
29:          Probability of pulse rate: $q_i = \frac{1}{\beta_i}$
30:          **if** $p_i > q_i$ **then**
31:             $x_i \leftarrow x_i + \gamma(x_i^* - x_i)$
32:          **else**
33:             $x_i \leftarrow x_i - \gamma(x_i - x_{best})$
34:          **end if**
35:       **end for**
36:       $t \leftarrow t + 1$
37:    **end while**
38:    $x^* \leftarrow x_{best}$
39:    $f^* \leftarrow f(x_{best})$
40: **end Function**

---

### 2.4. Grey Wolf Optimizer (GWO)

The Grey Wolf Optimizer (GWO) is an SI algorithm inspired by the hunting behavior of grey wolves in nature. The GWO imitates the social hierarchy and cooperative hunting strategies observed in wolf packs to guide the search for optimal solutions [9].

In the GWO, a population of candidate solutions, represented as grey wolves, explores the search space by adjusting their positions and mimicking the hunting behaviors of alpha, beta, and delta wolves. GWO implementation is shown in Algorithm 4 [14].

---

**Algorithm 4** GWO Pseudocode

---

**Require:**

  1: $n$: number of wolves

  2: $d$: dimension of the search space

  3: $t_{max}$: maximum number of iterations

  4: $a$: alpha coefficient

  5: $b$: beta coefficient

  6: $c$: delta coefficient

  7: $x_i$: position of wolf $i$

  8: $f(x_i)$: fitness of wolf $i$

  9: $x_i^*$: best position of wolf $i$

10: $f(x_i^*)$: best fitness of wolf $i$

**Ensure:**

11: $x^*$: optimal solution

12: **Function** GWO( )

13:    **for** $i = 1$ to $n$ **do**

14:       Initialize $x_i$ randomly

15:    **end for**

16:    **while** $t < t_{max}$ **do**

17:       **for** $i = 1$ to $n$ **do**

18:          Calculate $a$, $b$, and $c$

19:          Update position: $x_i \leftarrow x_i + a(x_i^* - x_i) + b(x_i^b - x_i) + c(x_i^c - x_i)$

20:       **end for**

21:       **for** $i = 1$ to $n$ **do**

22:          **if** $f(x_i) < f(x_i^*)$ **then**

23:             $x_i^* \leftarrow x_i$

24:          **end if**

25:       **end for**

26:       $t \leftarrow t + 1$

27:    **end while**

28:    $x^* \leftarrow x_{best}$

29:    $f^* \leftarrow f(x_{best})$

30: **end Function**

---

*2.5. Moth Flame Optimization (MFO)*

    The Moth Flame Optimization (MFO) is an SI algorithm inspired by the navigation behavior of moths in nature. The MFO mimics the attraction of moths toward artificial light sources to guide the search for optimal solutions [10]. Algorithm 5 displays MFO implementation [15].

---

**Algorithm 5** MFO Pseudocode

---

**Require:**
  1: $n$: number of moths
  2: $d$: dimension of the search space
  3: $t_{max}$: maximum number of iterations
  4: $a$: absorption coefficient
  5: $r$: random number
  6: $x_i$: position of moth $i$
  7: $f(x_i)$: fitness of moth $i$
  8: $x_i^*$: best position of moth $i$
  9: $f(x_i^*)$: best fitness of moth $i$
**Ensure:**
10: $x^*$: optimal solution
11: **Function** MFO()
12:   **for** $i = 1$ to $n$ **do**
13:     Initialize $x_i$ randomly
14:   **end for**
15:   **while** $t < t_{max}$ **do**
16:     **for** $i = 1$ to $n$ **do**
17:       Generate a new position $x_i'$
18:       Calculate the fitness of $x_i'$
19:       **if** $f(x_i') < f(x_i)$ **then**
20:         $x_i \leftarrow x_i'$
21:       **end if**
22:       Absorption: $x_i \leftarrow x_i - a(x_i^* - x_i)$
23:       Random walk: $x_i \leftarrow x_i + r(x_{best} - x_i)$
24:     **end for**
25:     $t \leftarrow t + 1$
26:   **end while**
27:   $x^* \leftarrow x_{best}$
28:   $f^* \leftarrow f(x_{best})$
29: **end Function**

---

## 3. Decentralized Neural Block Control (DNBC-UKF)

This section shows the proposed SI optimization approach for UKF learning of decentralized neural block control (DNBC-UKF) [16] applied to a 2-DOF robot manipulator.

For this purpose, we take the system to the following form

$$
\mathcal{X}^1_{i,k+1} = f^1_i\left(\mathcal{X}^1_i\right) + B^1_i\left(\mathcal{X}^1_i\right)\mathcal{X}^2_i + \Gamma^1_{i\ell},
$$

$$
\vdots \tag{1}
$$

$$
\mathcal{X}^r_{i,k+1} = f^r_i\left(\mathcal{X}^1_i, \cdots, \mathcal{X}^r_i\right) + B^r_i\left(\mathcal{X}^1_i, \cdots, \mathcal{X}^j_i\right)u_i + \Gamma^r_{i\ell}
$$

where $i = 1, \ldots, N$, $j = 1, \ldots, r-1$, $l = 1, \ldots, m_{ij}$. $N$ is the number of subsystems and $u_i \in \mathbb{R}^{m_i}$ is the input vector. $f^j_i$, $B^j_i$, and $\Gamma^j_i$ are assumed smooth and bounded functions, with $f^j_i(0) = 0$, and $B^j_i(0) = 0$; in addition, the structures of the subsystems are expressed by $m_{i1} \leq m_{i2} \leq \cdots \leq m_{ij} \leq p_i$. On the other hand, the interconnection terms $\Gamma^j_i$ are described by reflecting the relation between the $i$-th subsystem and the other ones.

The following RHONN structure is used in order to identify the behavior of system (1)

$$x_{i,k+1}^1 = w_{i,k}^1 S\left(\boldsymbol{\mathcal{X}}_{i,k}^1\right) + w_i^{'1} \boldsymbol{\mathcal{X}}_{i,k}^2,$$

$$\vdots \tag{2}$$

$$x_{i,k+1}^r = w_{i,k}^r S\left(\boldsymbol{\mathcal{X}}_{i,k}^1, \cdots, \boldsymbol{\mathcal{X}}_{i,k}^r\right) + w_i^{'r} \boldsymbol{u}_{i,k}$$

where $x_{i,k+1}^j = \begin{bmatrix} x_i^1 & x_i^2 & \cdots & x_i^r \end{bmatrix}^\top$ is the $j$-th block neuron state with $i = 1, \ldots, N$ and $j = 1, \ldots, r-1$; $w_{i,k}^{'j}$ are fixed parameters with $rank(w_i^{'j}) = m_{ij}$. $S(\bullet)$ is the activation function and $\boldsymbol{u}_{i,k}$ represents the input vector.

The NN training task consists of finding values of $w_i^{j,k}$ that minimize the identification error. For this reason, we propose to use a learning method using only the identification error information, such as the UKF described in Figure 2.

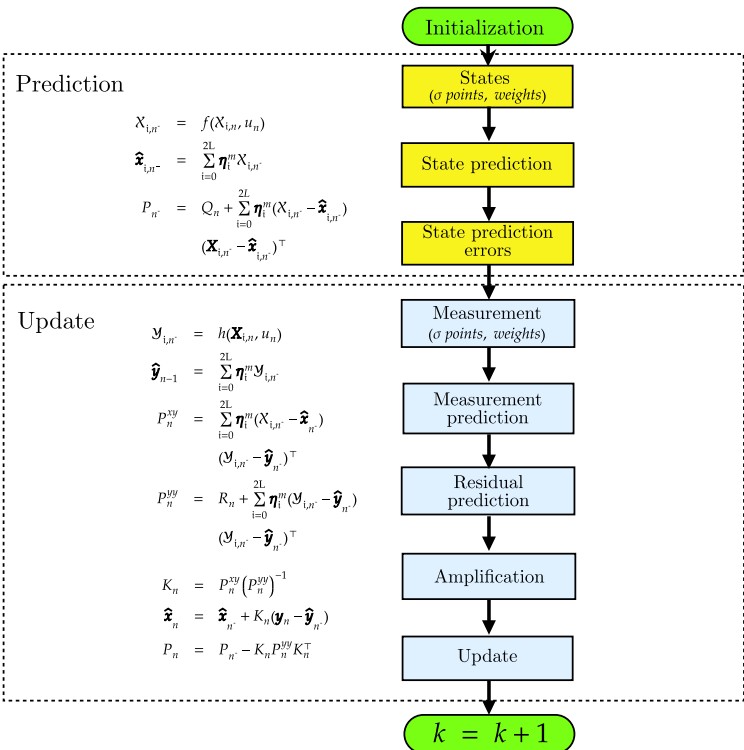

**Figure 2.** UKF framework.

The UKF is a powerful estimation and control tool: with wide applications in control theory; for the identification of nonlinear systems; and for the training of neural networks. Its versatility lies in its ability to handle nonlinear dynamics, non-Gaussian distributions, and uncertainties associated with real-world systems.

In the context of control theory, the UKF serves as an efficient means for state estimation in nonlinear systems. It allows for the real-time estimation of the system's internal states, which are often unobservable or difficult to measure directly [17]. By incorporating nonlinear models and the measured system outputs, the UKF provides accurate and reliable estimates of the system's states, enabling effective control strategies to be devised. The estimated states obtained from the UKF can then be utilized for feedback control, trajectory tracking, and system stabilization in a wide range of dynamic systems [18,19].

The UKF also plays a significant role in the identification of nonlinear systems. Identification refers to the process of determining the mathematical models or parameters that represent the underlying dynamics of a system based on observed input–output data. Non-

linear systems pose significant challenges in identification due to their complex dynamics. The UKF addresses these challenges by iteratively updating the system model parameters, enabling an accurate estimation of the nonlinear system's behavior. By leveraging the filtering and estimation capabilities of the UKF, researchers and engineers can effectively identify the dynamics, parameters, and structure of complex nonlinear systems, leading to an improved understanding and control of such systems [20].

Additionally, the UKF is employed in the training of neural networks, specifically in the context of Recurrent Neural Networks (RNNs). RNNs are powerful architectures for modeling sequential data and time series. It is possible to use the UKF in the training process to optimize the internal states, weights, and biases of the network in order to improve its learning capability and prediction accuracy [21]. By incorporating the UKF within the training process, the neural network can effectively capture and model the complex nonlinear dependencies present in the data, leading to an improved performance in tasks such as time series forecasting, speech recognition, and natural language processing [22,23].

The initialization of a UKF involves determining the initial state estimate, the covariance matrix and the process noise covariance matrix, which can make the selection of these initialization parameters a complex task, especially when using heuristic techniques [18]. The challenge arises because these parameters significantly impact the filter's performance and are often problem-specific, requiring domain expertise and careful tuning. The manual selection of these parameters can be time-consuming, and may not guarantee optimal performance.

In such scenarios, employing metaheuristic optimization methods proves to be a promising approach for selecting these initialization parameters. Metaheuristic optimization methods offer several advantages when applied to the selection of UKF initialization parameters. Firstly, these methods provide a systematic and automated approach to parameter tuning, relieving the burden of manual parameter selection. They can efficiently explore the vast parameter space, searching for the optimal combination that minimizes the error or maximizes a performance metric. By leveraging the search mechanisms inherent in metaheuristic algorithms, such as exploration and exploitation, the initialization parameters can be fine-tuned to enhance the convergence and accuracy of the UKF.

Secondly, metaheuristic optimization methods can handle nonlinearity, multimodality, and non-convexity in the optimization landscape, which are common challenges in parameter selection for UKF initialization. These algorithms possess the flexibility to adapt and explore diverse regions of the parameter space, avoiding local optima and finding near-optimal or globally optimal solutions.

### 3.1. SI Optimization for UKF Learning

Using the UKF to estimate the NN weights, and correcting for identification errors, the filter is updated at each step. Usually, $P_i^j$, $Q_i^j$, and $R_i^j$ are initialized as diagonal matrices with entries $P_i^j(0)$, $Q_i^j(0)$, and $R_i^j(0)$, respectively. Given that, typically, these entries are defined heuristically, we propose employing SI methods to improve the UKF training algorithm.

According to the optimal control theory [24], it is common to use error-based performance measures such as those described in Table 1. On the other hand, in (3), the Bolza form [25] is described and used as an objective function to evaluate the overall performance, including information from the control input of the system.

$$J = \underbrace{\underbrace{\mathbf{e}_{kf}^{\top} L \mathbf{e}_{kf} T}_{\text{Mayer form}} + \underbrace{\sum_{k_0}^{k_f} \left[ \mathbf{e}_k^{\top} Q \mathbf{e}_k + \mathbf{u}_k^{\top} R \mathbf{u}_k \right] T}_{\text{Lagrange form}}}_{\text{Bolza form}} \tag{3}$$

where $k_0$ is the initial iteration; $k_f$ is the final iteration; $T$ is the sampling time; $\mathbf{e}_k$ is the error vector; $\mathbf{u}_k$ is the control input vector; and $L$, $Q$, and $R$ are gain matrices with appropriate dimensions.

**Table 1.** Performance criteria.

| Criteria | Formula |
|---|---|
| Integral Absolute Error (IAE) | $\sum\limits_{k=0}^{\frac{t}{T}} \lvert\mathbf{e}_k\rvert$ |
| Integral Squared Error (ISE) | $\sum\limits_{k=0}^{\frac{t}{T}} \mathbf{e}_k^2$ |
| Integral Time-weighted Absolute Error (ITAE) | $\sum\limits_{k=0}^{\frac{t}{T}} k\lvert\mathbf{e}_k\rvert$ |
| Integral Time-weighted Squared Error (ITSE) | $\sum\limits_{k=0}^{\frac{t}{T}} k\mathbf{e}_k^2$ |

Although the Bolza form is a good performance criterion, it presents a serious disadvantage for this work. The final value of the error vector is not very useful for our methodology because it does not significantly represent the system identification and trajectory tracking; in other words, we need to know how it behaves throughout the simulation. For this reason, we propose an objective function, based on that found in [26], for the SI algorithms as follows:

$$f_{obj} = \varrho_1 \text{MSE}(e_{i,k}) + \varrho_2 \text{MSE}(z_{i,k}) + \sum_{k=0}^{\frac{t}{T}} [\varrho_3 (u_{i,k-1} - u_{i,k})] \tag{4}$$

where MSE represents the mean square error; $t$ is the total time of the simulation; $e_{i,k}$ represents the identification error; $z_{i,k}$ is the tracking error; $u_{i,k}$ represents the input control; and $\varrho_1$, $\varrho_2$, and $\varrho_3$ are scaling factors to bring all the terms of the objective function to a similar order.

*3.2. DNBC-UKF Controller Design*

Once the RHONN training has been defined, we design a controller based on the tracking error $z_i$ as follows:

$$\mathbf{z}_{\mathrm{i},k}^j = \mathbf{x}_{\mathrm{i},k}^j - \mathbf{x}_{\mathrm{id},k}^{1j} \tag{5}$$

where $\mathbf{x}_{\mathrm{id},k}^j$ is the desired trajectory signal and $\mathbf{x}_{\mathrm{i},k}^j$ is the NN state [27].

The new value is obtained as:

$$\mathbf{z}_{\mathrm{i},k+1}^j = \mathbf{w}_{\mathrm{i},k}^j S(\boldsymbol{\mathcal{X}}_{\mathrm{i},k}^1, \dots, \boldsymbol{\mathcal{X}}_{\mathrm{i},k}^j) + \mathbf{w}_{\mathrm{i}}^{'j} \mathbf{u}_{\mathrm{i},k} - \mathbf{x}_{\mathrm{id},k+1}^j. \tag{6}$$

Then, system (2) should be expressed as a function of variables $\mathbf{z}_{\mathrm{i},k}^j$ as:

$$\mathbf{z}_{\mathrm{i},k+1}^j = k_{\mathrm{i}}^j \mathbf{z}_{\mathrm{i},k}^j + \mathbf{w}_{\mathrm{i}}^{'j} \mathbf{u}_{\mathrm{i},k} - \mathbf{x}_{\mathrm{id},k+1}^j \tag{7}$$

When a sliding mode control strategy is implemented, the control input must be limited by $u_{0i}$ as:

$$\lvert \mathbf{u}_{\mathrm{i},k} \rvert \leq \mathbf{u}_{0i}. \tag{8}$$

The sliding surface is designed as $S_{D_{\mathrm{i}},k} = \mathbf{z}_{\mathrm{i},k}^r = 0$; then, system (7) is rewritten as follows:

$$S_{D_{\mathrm{i}},k+1} = \mathbf{w}_{\mathrm{i},k}^r S(\boldsymbol{\mathcal{X}}_{\mathrm{i},k}^1, \dots, \boldsymbol{\mathcal{X}}_{\mathrm{i},k}^r) + \mathbf{w}_{\mathrm{i}}^{'r} \mathbf{u}_{\mathrm{i},k} - \mathbf{x}_{\mathrm{id},k+1}^r. \tag{9}$$

The proper selection of the sliding manifold [28] presents the possibility of finding a bounded control law by $u_{0i}$; the control $u_{i,k}$ is composed as

$$
u_{i,k} = \begin{cases} u_{eq_i,k} & \text{for } \left\| u_{eq_i,k} \right\| \leq u_{0i}, \\[2ex] u_{0i} \dfrac{u_{eq_i,k}}{\left\| u_{eq_i,k} \right\|} & \text{for } \left\| u_{eq_i,k} \right\| > u_{0i}, \end{cases} \tag{10}
$$

where $u_{eq_i,k}$ is calculated from $S_{D_i,k+1} = 0$ as

$$
u_{eq_i,k} = \frac{1}{w_i'^r} \left[ -w_{i,k}^r S(\mathcal{X}_{i,k}^1, \dots, \mathcal{X}_{i,k}^r) + x_{id,k+1}^r \right]. \tag{11}
$$

Figure 3 illustrates the block diagram of the proposed SI optimization approach.

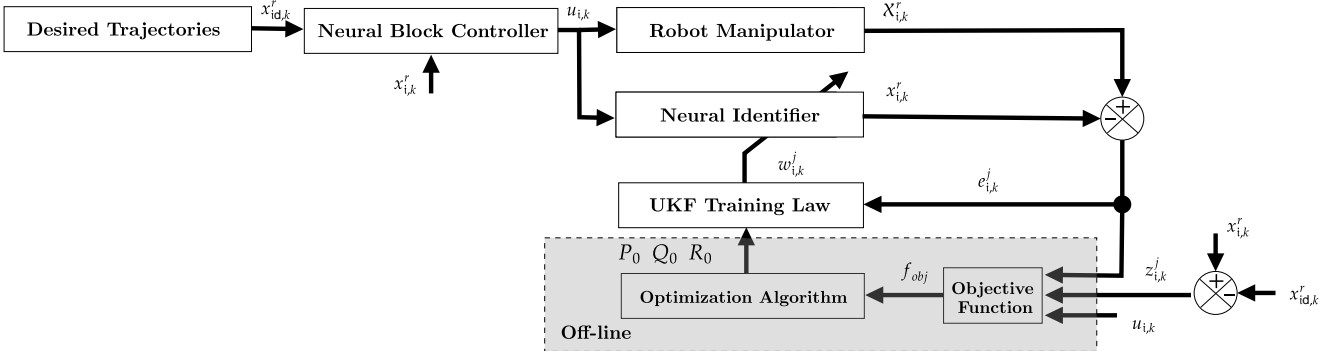

**Figure 3.** Decentralized neural identification and control scheme with SI optimization approach.

## 4. Results

The performance of the proposed approach was analyzed and compared using the following SI algorithms: ABC, ACO, ALO, BA, CS, GWO, MFO, PSO, APSO, and WOA. The comparisons were performed to find the algorithm that best minimizes the trajectory tracking error without exceeding the limits of the input torques.

This last condition of not surpassing the torque bounds is necessary for algorithm selection in real-time experiments. The experiments were performed on a 2-DOF vertical direct-drive robot manipulator, which is located at the Tecnologico Nacional de Mexico/Instituto Tecnologico de La Laguna, Mexico.

### 4.1. Prototype Description

To illustrate the implementation of the proposed scheme, we used the robot manipulator shown in Figure 4, which consists of two rigid links articulated by high-torque brushless direct-drive servos that present a reduced backlash and a significantly lower joint friction to drive the joints. The robot actuators act as torque sources and receive analog voltage as a torque reference signal. Joint positions are obtained using incremental encoders that send information to a DAQ [16].

The numerical values for the 2-DOF robot manipulator parameters alongside the dynamic model can be found in [29].

In order to prove the proposed approach, the discrete-time trajectories [27] were chosen as

$$
x_{1d,k}^1 = b_1(1 - e^{d_1 kT^3}) + c_1(1 - e^{d_1 kT^3})\sin(\omega_1 kT)[\text{rad}],
$$
$$
x_{2d,k}^1 = b_2(1 - e^{d_2 kT^3}) + c_2(1 - e^{d_2 kT^3})\sin(\omega_2 kT)[\text{rad}]
$$

where $b_1 = \pi/4$, $c_1 = \pi/18$, $d_1 = -2.0$, and $\omega_1 = 5$ [rad/s] are used for the first joint, while $b_2 = \pi/3$, $c_2 = 25\pi/36$, $d_2 = -1.8$, and $\omega_2 = 1$ [rad/s] are used for the second joint.

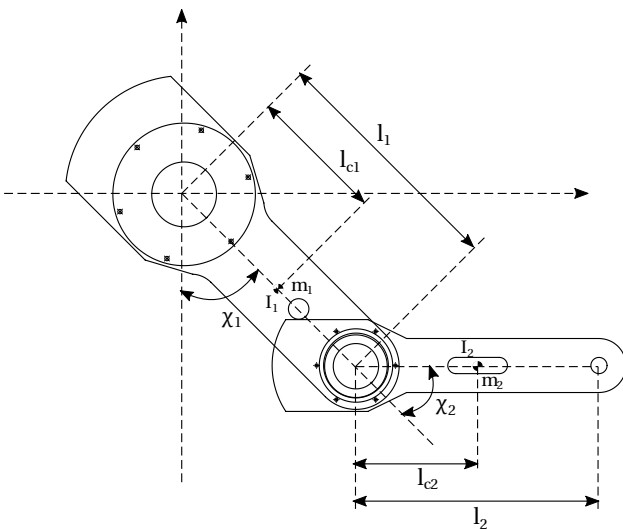

**Figure 4.** Diagram of the 2-DOF robot manipulator.

### 4.2. Simulation Results

For the simulations, the parameter settings of the SI algorithms considered are summarized as follows: starting with the common parameters, the number of iterations, which was 15; the population size was 30; the variables were 54; and the simulation time was 10 s. The particular parameter settings are given in Table 2.

**Table 2.** Parameter settings for SI algorithms.

| Reference | Algorithm | Parameter Values |
|:---:|:---:|:---:|
| [30] | ABC | Limit: 100, $F_l$: 0.1, $F_u$: 0.9, $p$: 0.5 |
| [31] | ACO | $\alpha$: 1.0, $\beta$: 3.0, Evaporation Rate: 0.5 |
| [32] | ALO | Probabilistic Switch: 0.1, Random Walk Length: 1.5, Levy Flight $a$: 1.0, $b$: 1.0 |
| [33] | BA | $A$: 0.25, $r$: 0.5, $\alpha$: 1.0, $\gamma$: 0.1, $f_{min}$: 0.0, $f_{max}$: 2.0 |
| [34] | CS | Discover Rate $p_a$: 0.25, Levy Flight $a$: 0.1, $b$: 0.9 |
| [35] | GWO | $a_0$: 2.0 |
| [36] | MFO | $a$: 1.0, $b$: 1.0 |
| [37] | PSO | $\varphi_1$: 2.0, $\varphi_1$: 2.0, $w$: 0.7, $v_{max}$: 0.1 |
| [37] | APSO | $\varphi_1$: 1.5, $\varphi_2$: 1.5, $w$: 0.7, $v_{max}$: 0.1, $p_a$: 0.1, $p_r$: 0.1 |
| [38] | WOA | $a_1$: 2.0, $a_2$: $-1.0$ |

SI algorithms presented in Table 2 were considered for comparison purposes because they have been employed previously in the state-of-the-art for neural network training, as reported in [39–48].

In all simulations, the specifications of the test machine were an AMD Ryzen 9 4900HS® (AMD Ryzen is a registered trademark of Advanced Micro Devices, Inc., Santa Clara, CA, USA ) CPU 3.0 GHz and 16 GB of RAM. Moreover, the experiments were performed in the MATLAB R2015a environment® (MATLAB is a registered trademark of MathWorks, Inc., Natick, MA, USA).

For comparative purposes, we tested each SI algorithm 50 times independently. To qualify the results, we calculated statistical data of the mean, standard deviation (SD), and the best and worst results for different performance indices and the proposed objective function. The performance of the algorithms is reflected in a small mean value with a low standard distribution, implying a small difference between the best and worst results. Table 3 shows the performance measures.

**Table 3.** Simulation results performance.

| | IAE | ISE | ITAE | ITSE | BOLZA | $F_{obj}$ | | IAE | ISE | ITAE | ITSE | BOLZA | $F_{obj}$ |
|---|---|---|---|---|---|---|---|---|---|---|---|---|---|
| | | | **ABC** | | | | | | | **ACO** | | | |
| Best | 1.6659 | 2.0687 | 1.9251 | 3.1455 | 1.2122 | 1.4809 | Best | 1.1065 | 7.0311 | 1.9234 | 0.7806 | 1.3876 | 1.7014 |
| Worst | 4.9574 | 17.3234 | 4.4748 | 14.6351 | 2.9154 | 2.5511 | Worst | 8.4880 | 55.0904 | 3.7347 | 7.7709 | 4.0725 | 2.8205 |
| Mean | 3.0176 | 8.3139 | 3.3381 | 6.4803 | 1.9728 | 1.7846 | Mean | 2.1222 | 19.0324 | 2.4459 | 3.2221 | 2.7406 | 1.9049 |
| SD | 0.6569 | 3.5723 | 0.4139 | 1.9143 | 0.2121 | 0.0855 | SD | 0.5998 | 6.9280 | 0.3136 | 0.8942 | 0.4055 | 0.1137 |
| Avg. CPU time | 385.6374 | 382.4414 | 377.5315 | 388.1568 | 329.2899 | 327.3551 | Avg. CPU time | 319.4551 | 270.9993 | 273.8600 | 285.6364 | 271.8496 | 271.7896 |
| | | | **ALO** | | | | | | | **BA** | | | |
| Best | 0.4117 | 0.8757 | 0.1221 | 0.0297 | 0.4798 | 0.7722 | Best | 0.4622 | 0.6824 | 0.2282 | 0.0778 | 0.8319 | 0.7865 |
| Worst | 2.4396 | 8.0046 | 1.6833 | 1.7144 | 2.3748 | 2.4519 | Worst | 3.0243 | 6.5020 | 1.7291 | 5.6521 | 2.5186 | 2.1805 |
| Mean | 0.9106 | 2.1424 | 0.5598 | 0.4147 | 0.9967 | 1.4478 | Mean | 0.9396 | 2.2031 | 0.6312 | 1.6451 | 1.4602 | 1.5367 |
| SD | 0.4119 | 0.9296 | 0.3396 | 0.3418 | 0.4452 | 0.1835 | SD | 0.4535 | 1.3601 | 0.3055 | 1.2372 | 0.2681 | 0.1425 |
| Avg. CPU time | 195.2782 | 193.8541 | 188.2104 | 195.6159 | 199.8425 | 193.8247 | Avg. CPU time | 162.2854 | 251.0991 | 173.2529 | 172.2656 | 234.0957 | 188.7852 |
| | | | **CS** | | | | | | | **GWO** | | | |
| Best | 1.1682 | 8.6384 | 1.9009 | 2.2488 | 2.1922 | 2.2349 | Best | 0.5485 | 0.8409 | 0.2685 | 0.1061 | 0.9654 | 1.0775 |
| Worst | 7.6640 | 26.8575 | 5.1201 | 17.8572 | 4.8216 | 3.7666 | Worst | 4.0810 | 13.8469 | 6.0435 | 11.2769 | 4.6651 | 2.7274 |
| Mean | 3.3023 | 17.9493 | 2.9659 | 6.2737 | 3.5290 | 2.9180 | Mean | 1.1522 | 2.2295 | 2.2620 | 1.5815 | 2.1502 | 1.7703 |
| SD | 1.2205 | 3.2834 | 0.7816 | 2.6353 | 0.4599 | 0.2444 | SD | 0.8178 | 2.8493 | 1.0776 | 1.8125 | 0.8949 | 0.2275 |
| Avg. CPU time | 317.7669 | 313.6375 | 312.7115 | 313.0690 | 312.2985 | 313.7871 | Avg. CPU time | 197.7674 | 199.4584 | 200.7393 | 200.1622 | 225.8913 | 199.8676 |
| | | | **MFO** | | | | | | | **PSO** | | | |
| Best | 0.5013 | 0.6742 | 0.2346 | 0.0871 | 0.6720 | 1.0254 | Best | 1.5053 | 4.6942 | 0.5468 | 1.5487 | 1.4258 | 1.4388 |
| Worst | 4.4130 | 10.5064 | 5.7266 | 7.2781 | 4.6758 | 2.7274 | Worst | 6.4121 | 29.8629 | 3.3398 | 11.8439 | 4.6686 | 3.8913 |
| Mean | 2.0395 | 3.5393 | 2.7144 | 1.1115 | 2.8113 | 1.7599 | Mean | 2.5117 | 14.8761 | 2.2052 | 5.0699 | 2.3209 | 2.1410 |
| SD | 0.9656 | 1.8749 | 1.1719 | 1.4426 | 0.9352 | 0.2437 | SD | 0.7388 | 6.2862 | 0.3225 | 2.2650 | 0.7198 | 0.4797 |
| Avg. CPU time | 196.3359 | 193.7482 | 196.1711 | 195.6669 | 194.9560 | 195.0981 | Avg. CPU time | 205.5427 | 204.3713 | 205.1566 | 204.0309 | 204.3103 | 203.5299 |

**Table 3.** *Cont.*

| | APSO | | | | | | | WOA | | | | | |
| | IAE | ISE | ITAE | ITSE | BOLZA | $F_{obj}$ | | IAE | ISE | ITAE | ITSE | BOLZA | $F_{obj}$ |
|---|---|---|---|---|---|---|---|---|---|---|---|---|---|
| **Best** | 1.2219 | 9.6568 | 1.0353 | 2.0965 | 1.3454 | 1.3019 | **Best** | 0.7574 | 0.9207 | 0.9669 | 0.3357 | 1.0693 | 1.3655 |
| **Worst** | 7.5709 | 89.5887 | 5.9703 | 21.9045 | 12.5607 | 5.9705 | **Worst** | 4.3771 | 45.0806 | 4.9312 | 8.2819 | 4.7169 | 4.7380 |
| **Mean** | 3.2289 | 36.6708 | 2.3626 | 11.7262 | 2.2305 | 3.4699 | **Mean** | 2.0837 | 2.5373 | 1.8365 | 1.9530 | 2.3689 | 1.9793 |
| **SD** | 0.9328 | 15.0780 | 0.7621 | 3.7311 | 1.2944 | 0.8019 | **SD** | 0.5674 | 3.6982 | 0.6706 | 1.2024 | 0.5636 | 0.3771 |
| **Avg. CPU time** | 195.5655 | 194.8540 | 195.7047 | 195.3598 | 228.1618 | 203.4104 | **Avg. CPU time** | 171.9729 | 163.0374 | 163.3186 | 162.5492 | 155.6390 | 155.6390 |

*4.3. Experimental Results*

The selection of the ALO, BA, GWO, MFO, and PSO algorithms for real-time experiments was because they show a balanced performance with low computational costs. However, the main reason was that during the entire simulation, none of them exceeded the torque limits.

Real-time experiments were implemented using Ansi C on WinMechLab, a real-time platform running on an Intel Pentium 4 PC with real-time Windows XP, with a 2.5 ms sampling period, and using a MultiQ-PCI data acquisition board from Quanser Consulting Inc., Markham, ON, CAN [49].

Figures 5 and 6 show the tracking trajectories for each link obtained in the simulation using the SI algorithms compared to the non-optimized UKF, we include in Table 4 and Figure 7 the values of the performance of the algorithms by evaluating the objective function and $\mathcal{L}_2$-norm, described by

$$\mathcal{L}_2\text{-norm} = \sqrt{\frac{T}{t} \sum_{i=0}^{n} ||z_{i,k}||^2}$$

where $T$ is the sampling time and $t$ is the total time of the simulation, which, for this case is 20 s. In addition, Table 5 includes the RMS of the joint input torques comparing each algorithm with the UKF with non-optimized parameters. Moreover, the input pairs for each link are shown in Figures 8a to 9e.

**Table 4.** Objective function evaluation for real-time experimentation.

| Algorithm UKF | $f_{obj}$ |
|---|---|
| non-optimized | 2.0023 |
| ALO | 1.2678 |
| BA | 1.3529 |
| GWO | 1.4797 |
| MFO | 1.3948 |
| PSO | 1.7829 |

**Table 5.** RMS for joint input torques of SI algorithms.

| Algorithm UKF | RMS($u_{1,k}$) | RMS($u_{2,k}$) |
|---|---|---|
| non-optimized | 30.6562 | 2.5715 |
| ALO | 30.5262 | 2.5931 |
| BA | 30.3053 | 2.6880 |
| GWO | 30.7568 | 3.2744 |
| MFO | 30.3398 | 2.8736 |
| PSO | 30.5262 | 2.5931 |

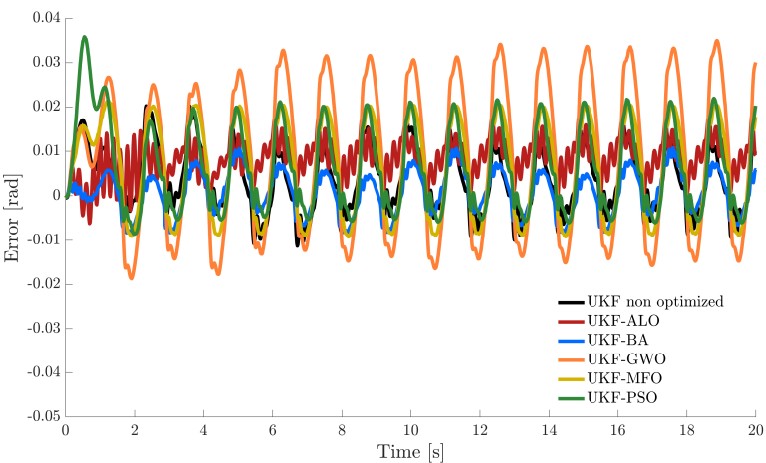

**Figure 5.** Trajectory tracking position error link 1.

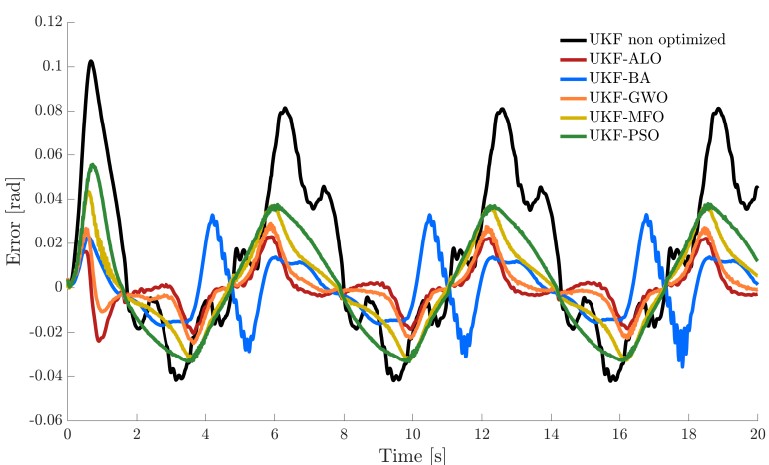

**Figure 6.** Trajectory tracking position error link 2.

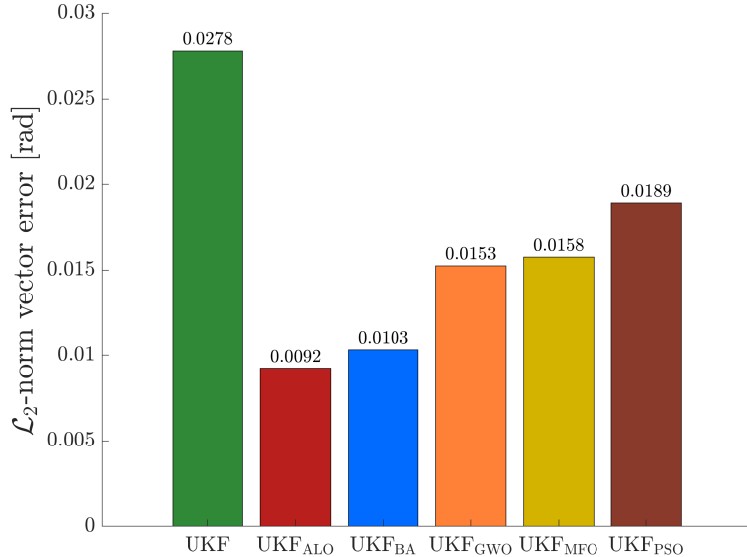

**Figure 7.** $\mathcal{L}_2$-norm for SI algorithms.

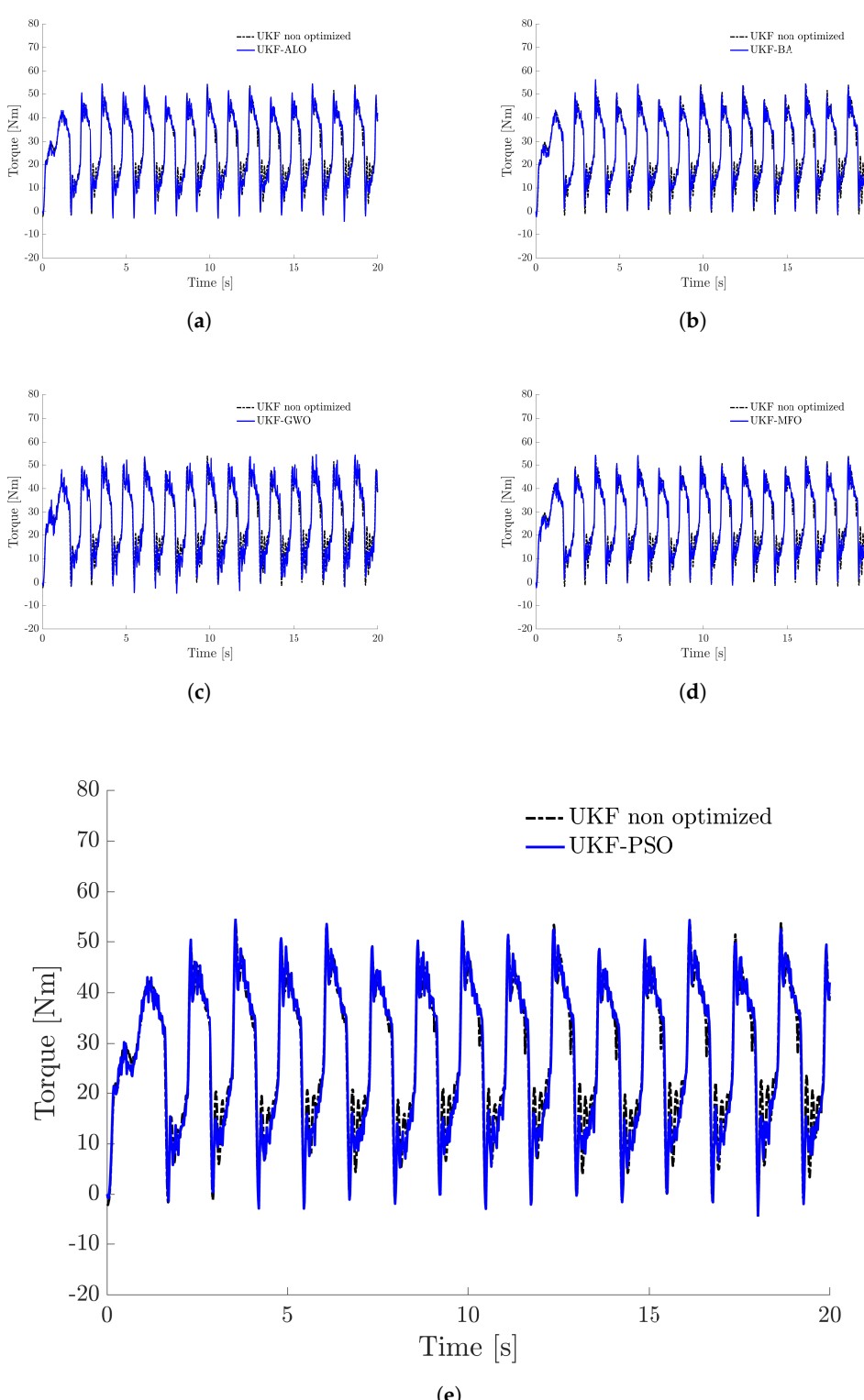

**Figure 8.** Input torque link 1 with SI optimization. (**a**) Input torque link 1 ALO. (**b**) Input torque link 1 BA. (**c**) Input torque link 1 GWO. (**d**) Input torque link 1 MFO. (**e**) Input torque link 1 PSO.

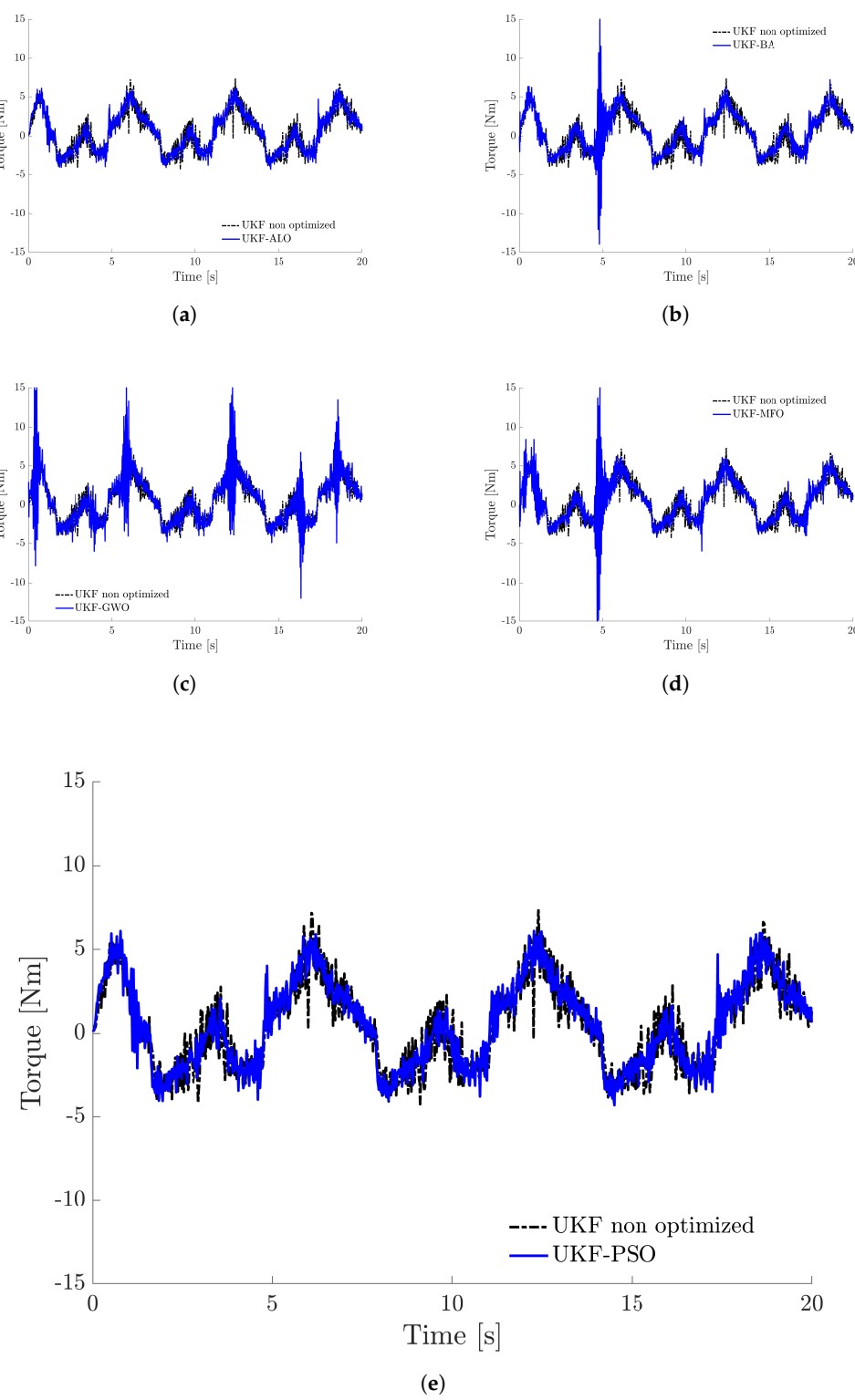

**Figure 9.** Input torque link 2 with SI optimization. (**a**) Input torque link 2 ALO. (**b**) Input torque link 2 BA. (**c**) Input torque link 2 GWO. (**d**) Input torque link 2 MFO. (**e**) Input torque link 2 PSO.

## 5. Discussion

For the tracking control problem of a 2-DOF robot manipulator, we proposed a DNBC controller, which does not require any knowledge of the system since it uses a UKF-trained neural identifier, whose added value presents an optimization tuning via SI algorithms. Everyone tuned the initial filter parameters in an off-line simulation subsequently used in real-time experiments.

The selection of SI algorithms employed in this work is due to the following criteria:

1. Few tuning parameters.
2. Good exploitation–exploration balance.
3. Low computational cost concerning their performance.

The computational cost is the factor that we consider to be most important when carrying out the proposed optimization task, at least for the off-line simulation. Since the stochastic nature of both the UKF filter and the algorithms must be taken into account, in addition to the added cost of the controller, this translates into an increase in the number of attempts to achieve acceptable results. For this reason, Table 3 shows the average CPU time for each SI algorithm utilized.

Interpreting the results obtained in the off-line simulation, we can highlight the following:

1. According to Tables 2 and 3, all algorithms meet the selection criteria, have few parameters, acceptable performance, and relatively low CPU time.
2. All performance indices show similar average CPU time for each algorithm. The algorithm with the lowest average CPU time was BA, while the highest was ABC.
3. Based on the statistical mean and standard deviation, the best-performing algorithm overall was ALO, followed by BA, GWO, and MFO, in that order.

In the case of real-time experiments, the principal selection criterion is to maintain joint torque limits. Only the Bolza form (3) and the objective function (4) of all performance measures consider the joint torques. Although Table 3 shows that all algorithms perform well, input torque signals in the simulation of the ABC, ACO, CS, APSO, and WOA algorithms exceeded the joint torque limits. For this reason, they were discarded from being used in real-time experiments.

Interpreting the results obtained in real-time, we can reach the following final remarks:

1. According to Figures 5–7, all algorithms perform better than the UKF without optimization. This demonstrates the advantages of using the proposed methodology. Table 4 shows the values of the objective function (4) evaluated in all used algorithms; the performance of these algorithms is reflected by minimizing the value of the objective function since it is described in terms of tracking and identification errors. As we can observe, the one with the best performance is the ALO, which we can also contrast in Figure 7. The controller performance shows a notorious improvement concerning a previous heuristic tuning.
2. Concerning Figure 5, the performance of the GWO for the first link is not up to par. However, according to Figure 6, this algorithm on the second link performs better than the other algorithms. Figure 9c exhibits this in the noise of the input torque signal, which is reflected in the RMS value in Table 5.

## 6. Conclusions

In this work, we have presented the implementation of SI-inspired algorithms in the selection of UKF initialization parameters and their real-time application in a discrete-time decentralized neural block control scheme. We proposed a new objective function that effectively utilizes information from trajectory tracking and identification errors paired with the slopes of the input torques. This function allows us to meet minimizing tracking errors without overshooting the bounds on the control input signals. We performed in simulation a comparative experimental study of the performance of the following SI algorithms: ABC, ACO, ALO, BA, CS, GWO, MFO, PSO, APSO, and WOA. For this analysis, we used five performance indices in addition to our proposed objective function. The real-time experiments were carried out on a 2-DOF robot manipulator, showing ALO, BA, GWO, MFO, and PSO performance, which were the best in our comparative study.

In summary, the main contribution of this work is the implementation of the use of SI-inspired algorithms in the selection of UKF initialization parameters and its real-time implementation in a discrete-time decentralized neural block control; moreover, an

experimental comparative study of performance was carried out between the ALO, BA, GWO, MFO, and PSO.

**Author Contributions:** Conceptualization, J.F.G., R.G.-H., M.A.L. and V.S; methodology, J.F.G., R.G.-H., M.A.L. and V.S; software, J.F.G. and V.S; validation, J.F.G., R.G.-H., M.A.L. and V.S.; formal analysis, J.F.G., R.G.-H., M.A.L. and V.S.; investigation, J.F.G., R.G.-H., M.A.L. and V.S.; resources, J.F.G., R.G.-H., M.A.L. and V.S.; writing—original draft preparation, J.F.G. and R.G.-H.; writing—review and editing, J.F.G., R.G.-H., M.A.L. and V.S.; visualization, J.F.G., R.G.-H., M.A.L. and V.S.; supervision, R.G.-H., M.A.L. and V.S.; project administration, R.G.-H.; funding acquisition, R.G.-H., M.A.L. and V.S. All authors have read and agreed to the published version of the manuscript.

**Funding:** This research was funded by financial support of the research projects of the Tecnológico Nacional de México/I. T. La Laguna and CONACYT.

**Data Availability Statement:** Data sharing not applicable.

**Acknowledgments:** The authors would like to thank to all the staff in División de Estudios de Posgrado e Investigación del Tecnológico Nacional de México/I. T. de La Laguna.

**Conflicts of Interest:** The authors declare no conflict of interest.

## Abbreviations

The following abbreviations are used in this manuscript:

| | |
|---|---|
| ABC | Artificial Bee Colony |
| ACO | Ant Colony Optimization |
| ALO | Ant Lion Optimizer |
| APSO | Accelerated Particle Swarm Optimization |
| BA | Bat Algorithm |
| CS | Cuckoo Search |
| DOF | Degrees of Freedom |
| EKF | Extended Kalman filter |
| GWO | Grey Wolf Optimizer |
| IAE | Integral Absolute Error |
| ISE | Integral Squared Error |
| ITAE | Integral Time-weighted Absolute Error |
| ITSE | Integral Time-weighted Squared Error |
| MFO | Moth Flame Optimization |
| MSE | Mean Square Error |
| NBC | Nonlinear Block Control |
| NN | Neural network |
| PSO | Particle Swarm Optimization |
| RHONN | Recurrent High Order Neural Networks |
| RMS | Root Mean Square |
| SD | Standard Deviation |
| SI | Swarm Intelligence |
| UKF | Unscented Kalman filter |
| WOA | Whale Optimization Algorithm |

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
