# Peer review of "A Comparative Study of Swarm Intelligence Metaheuristics in UKF-Based Neural Training Applied to the Identification and Control of Robotic Manipulator"

_algorithms, doi:10.3390/a16080393_

Round 1
Reviewer 1 Report
This work proposes a comprehensive comparative analysis of four prominent Swarm Intelligence (SI) optimization algorithms: Ant Lion Optimizer (ALO), Bat Algorithm (BA), Grey Wolf Optimizer (GWO), and Moth Flame Optimization (MFO). The algorithms are applied in real-time in the Decentralized Neural Block Control (DNBC) scheme for trajectory tracking control of a 2-DOF robot manipulator to analyze the performance difference of five algorithms. Although this paper has sufficient comparison of the algorithm, the following questions need to be solved:
1) Four algorithms are mentioned in the abstract, but five algorithms are compared in research, and PSO is not mentioned in the abstract.
2) A drawing error has occurred in Figure 3.
3) The overspent chapters in the second section illustrate the basic concepts of different algorithms and are not helpful to the article.
4) There are so many kinds of swarm intelligence algorithms, Why the five algorithms are selected for comparison is not clear to express in the article. What advantages do the five algorithms have over the other algorithms?
5) The algorithm parameters are one of the factors that affect the performance of the algorithm. Therefore, what method is the parameter setting in lines 317-326 or what is the basis for the determination?
6) To better observe the tracking error of different algorithms, it is suggested to put the six curves of the five graphs in the same graph.
7) Figure 5-Figure 8 has missing titles.
8) The references should be updated. Besides, some critical and closed related researches are missing.
Minor editing of English language required
Reviewer 2 Report
The authors proposed a comparative study of swarm intelligence metaheuristics for UKF-based neural training applied to the identification and control of the robotic manipulator. The job was interesting but the overall work should be improved before further operations.
1. Only four of the metaheuristics were not enough and could not represent the swarm intelligence.
2. More efforts should be involved in the literature review of the development of metaheuristics.
3. More simulation experiments should be carried out to compare and prove the efficiency.
4. Codes or pseudocodes should be shared or illustrated.
5. Conclusions should be written more and discussions should be made.
normal
Reviewer 3 Report
1. Too many pages are spent describing prior approaches and algorithms; these descriptions can be compressed. Figure 1, which merely shows four ellipses, is not a useful depiction of an algorithm or approach.
2. The introduction section should begin by motivating the problem addressed and outlining the approach used (instead of describing UKF).
3. The PSO algorithm described is a very old one--many new variations have been developed in the last three decades, improving the algorithm. Comparisons with one or two recent variants are needed, instead of the old PSO version.
4. Complexity comparisons can be misleading with metaheuristic algorithms: each iteration may take less time, but this may be balanced out by the need to execute many more iterations.
5. Several figures are presented showing visual performance comparisons which are hard for a reader to assess. Tabular numerical results may help better convey the results.
Reviewer 4 Report
The authors presented an approach to improve the Kalman Filter (Unscented Kalman Filter, UKF) as a result of optimization using four well-known metaheuristic algorithms. The results are interesting and promising. However, I have a few suggestions for improving the manuscript.
1. The authors explain the choice of four metaheuristic algorithms for research only in the conclusion. However, the question of the validity of such a choice will clearly arise for readers earlier. I propose to justify the choice of the algorithms under study in more detail in the introduction.
2. The authors presented the results of only a theoretical evaluation of the performance and complexity of the algorithms under study in section 2.6. Only the error rate of L2-norm algorithms has been studied in detail. I propose to show the results of a study of the computational performance of algorithms. Especially since the authors mention performance in the Сonclusion. In addition, I propose to detail the experimental setup, that is, the software and hardware used for the simulation.
3. The authors use the same objective function to evaluate the effectiveness of the developed algorithms, judging by their previous articles, for example [16]. However, a set of test functions is usually used to evaluate the performance and accuracy of optimization methods and algorithms. I propose to justify the choice of this specific objective function for the evaluation of algorithms.
4. Figure 1 is not very informative. I suggest replacing it with a flowchart or diagram of the research methodology.
5. The description of known heuristic algorithms is overly detailed. I propose to reduce this known information by replacing it with reviews of studies on the performance and efficiency of these algorithms.
6. On the other hand, the authors used many unsubstantiated statements in the introduction without the necessary references to supporting studies. I propose to supplement the References with articles on relevant research.
7. Remove the dot before [5] on line 32.
Round 2
Reviewer 1 Report
This paper proposed a comprehensive comparative analysis of four prominent Swarm Intelligence (SI) optimization algorithms. The revised version of the paper has made improvements to the issues raised. The following suggestions are listed to improve its presentation.
--Discuss the practical implications. In addition, future work needs to explained more in detail.
--This manuscript fails to support the novelty and the contribution of the work done.
--The paper needs editing for language and grammar.
The paper needs editing for language and grammar.
Reviewer 2 Report
The improved version of this paper is better now. but there is a strong recommendation of lists without any dot-signs.
The dot-signs or any other particular symbolic signs are usually used in PPT slides to illustrate the parallel items. Lists with numbers are recommended.
Try to revise them.
better and could be published now.
Reviewer 3 Report
As mentioned in my previous comment#2, the problem being addressed should be introduced very early, e.g., summarizing some of the text in Section 3.
My main remaining concern is about the variant of PSO (APSO) used for comparison. Note that it does not perform as well as the original PSO, and I believe that this particular variation is rarely used in practice. Also, in the PSO literature, "APSO" usually refers to "Adaptive PSO", which is more popular than "Accelerated PSO". I recommend trying another version of PSO which has been found to be more successful than the 1995 version: a simple suggestion is to select one which has received more citations in the literature, e.g., EPSO, proposed in:
Pugh J, Martinoli A (2007) Inspiring and modeling multi-robot search with particle swarm optimization. In: 2007 IEEE swarm intelligence symposium. IEEE, pp 332–339
Other citations can be found at a review article at https://link.springer.com/article/10.1007/s11831-022-09819-3
Even for the currently cited Accelerated PSO, the original authors who proposed the algorithm should be cited, not a recent application; Wikipedia gives the reference as: X. S. Yang, S. Deb and S. Fong, Accelerated particle swarm optimization and support vector machine for business optimization and applications, NDT 2011, Springer CCIS 136, pp. 53-66 (2011).
Also, in your paper, APSO is mentioned in this paper but not described.
Reviewer 4 Report
The authors have done significant work to improve the manuscript. However, I have minor new suggestions to improve the second version of the manuscript.
1. The criteria for choosing the algorithms under study should be moved from the Discussion section to the Introduction section. Moreover, the choice of these criteria should be proven by references to relevant studies.
2. The Discussion section should contain a comparison of the obtained results with previous studies.
3. The addition of nine references [39 - 48] in section 4.2 is belated and looks formal. I propose to analyze in detail the results of these studies in the introduction.
Round 3
Reviewer 3 Report
Comments from the previous review have been addressed satisfactorily.